# Inferring gender from first names: Comparing the accuracy of Genderize, Gender API, and the gender R package on authors of diverse nationality

Alexander D. VanHelene[1,2,3], Ishaani Khatri[4], C. Beau Hilton[5], Sanjay Mishra[1,2,4], Ece D. Gamsiz Uzun[2,4,6,7], Jeremy L. Warner[1,2,4,8,9]*

1 Lifespan Cancer Institute, Rhode Island Hospital, Providence, Rhode Island, United States of America, 2 Center for Clinical Cancer Informatics and Data Science, Legorreta Cancer Center, Brown University, Providence, Rhode Island, 3 Department of Biomedical Informatics, University of Pittsburgh, Pittsburgh, Pennsylvania, United States of America, 4 Warren Alpert Medical School, Brown University, Providence, Rhode Island, United States of America, 5 Department of Internal Medicine, Vanderbilt University, Nashville, Tennessee, United States of America, 6 Center for Computational Molecular Biology, Brown University, Providence, Rhode Island, United States of America, 7 Department of Pathology and Laboratory Medicine, Brown University, Providence, Rhode Island, United States of America, 8 Department of Medicine, Brown University, Providence, Rhode Island, United States of America, 9 Department of Biostatistics, Brown University, Providence, Rhode Island, United States of America

* jeremy_warner@brown.edu

## Abstract

Meta-researchers commonly leverage tools that infer gender from first names, especially when studying gender disparities. However, tools vary in their accuracy, ease of use, and cost. The objective of this study was to compare the accuracy and cost of the commercial software Genderize and Gender API, and the open-source gender R package. Differences in binary gender prediction accuracy between the three services were evaluated. Gender prediction accuracy was tested on a multi-national dataset of 32,968 gender-labeled clinical trial authors. Additionally, two datasets from previous studies with 5779 and 6131 names, respectively, were re-evaluated with modern implementations of Genderize and Gender API. The gender inference accuracy of Genderize and Gender API were compared, both with and without supplying trialists' country of origin in the API call. The accuracy of the gender R package was only evaluated without supplying countries of origin. The accuracy of Genderize, Gender API, and the gender R package were defined as the percentage of correct gender predictions. Accuracy differences between methods were evaluated using McNemar's test. Genderize and Gender API demonstrated 96.6% and 96.1% accuracy, respectively, when countries of origin were not supplied in the API calls. Genderize and Gender API achieved the highest accuracy when predicting the gender of German authors with accuracies greater than 98%. Genderize and Gender API were least accurate with South Korean, Chinese, Singaporean, and Taiwanese authors, demonstrating below 82% accuracy. Genderize can provide similar accuracy to Gender API while being 4.85x less expensive. The gender R package achieved below 86% accuracy on the full dataset. In the replication studies, Genderize and gender API demonstrated better performance than in the

**Data Availability Statement:** The data that support the findings of this study are publicly available from the Harvard DataVerse: https://dataverse.harvard.edu/dataset.xhtml?persistentId=doi:10.7910/DVN/HZSTFZ.

**Funding:** This work was supported by grants from the National Cancer Institute: U24 CA265879 and U24 CA248010 (https://www.cancer.gov). ADV, SM, and JLW were supported by U24 CA265879. ADV, EDGU, and JLW were supported by U24 CA248010. The funders had no role in study design, data collection and analysis, decision to publish, or preparation of the manuscript.

**Competing interests:** JLW is supported by grants from the National Cancer Institute (NCI), the American Association for Cancer Research (AACR), and Brown Physicians Incorporated through his institution. He reports consulting for Westat and The Lewin Group, outside the submitted work. He reports ownership of HemOnc.org LLC; shares have no monetary value. He is the Chief Technology Officer of HemOnc.org LLC; this position is uncompensated. SM is supported by grants from the NCI and the AACR through his institution. He has received money from the National Institutes of Health (NIH) and the Federation of American Societies for Experimental Biology (FASEB), personal fees from the Disney Corporation, and hospitality from the Schmidt Family Foundation. ADV, IK, CGH, and EGDU declare that no competing interests exist.

original publications. Our results indicate that Genderize and Gender API achieve similar accuracy on a multinational dataset. The gender R package is uniformly less accurate than Genderize and Gender API.

## Author summary

Gender disparities in academia have prompted researchers to investigate gender gaps in professorship roles and publication authorship. Of particular concern are the gender gaps in cancer clinical trial authorship. Methodologies that evaluate gender disparities in academia often rely on tools that infer gender from first names. Tools that predict gender from first names are often used in methodologies that determine the gender ratios of academic departments or publishing authors in a discipline. However, researchers must choose between different gender predicting tools that vary in their accuracy, ease of use, and cost. We evaluated the binary gender prediction accuracy of Genderize, Gender API, and the gender R package on a gold-standard dataset of 32,968 clinical trialists from around the world. Genderize and Gender API are commercially available, while the gender R package is free and open source. We found that Genderize and Gender API were more accurate than the gender R package. In addition, Genderize is cheaper than Gender API, but is more sensitive to inconsistencies in name formatting and the presence of diacritical marks. Both Genderize and Gender API were most accurate with non-Asian names.

## Introduction

One of the most well-documented disparities in STEM is gender disparity [1,2]. This issue is especially notable in the cancer clinical trial domain, with underrepresentation of women in the leadership of pivotal trials documented as recently as within the last decade [3]. Furthermore, gender disparities in healthcare accessibility have also been documented, and gender disparity also affects public health, policy-making, and diversity metrics [4]. The study of gender disparity in scientific authorship and other contexts often requires the determination of gender from very limited data, e.g., author forenames. Software [5–10] that infers gender from forenames could potentially enable researchers to automate gender prediction in large datasets. Commercial gender prediction services [11,12] such as Genderize and Gender API programmatically predict gender from first names. The gender R package [13] is an open-source alternative to these proprietary gender prediction tools.

Gender prediction software has demonstrated high accuracy when evaluating non-Asian first names, but often falters when evaluating names from Asian cultures [14]. Further, the presence of diacritical marks and hyphens reportedly affects the accuracy of gender prediction in some tools [15]. Few studies [16] to date have evaluated differences in accuracy in gender predicting software between non-Asian and Asian names. To our knowledge, no studies have evaluated how different ways of delimiting two-part first names e.g. Jean-Pierre vs Jean Pierre vs Jeanpierre, affect gender prediction accuracy.

We compared the gender prediction accuracy of Genderize, Gender API, and the gender R package using a large manually curated registry of cancer clinical trialists with labeled genders and diverse nationalities. In addition, we quantified the accuracy of these tools by author nationality and compared different strategies for delimiting two-part forenames, which are

common in the English language spelling of Korean, Chinese, Singaporean, and Taiwanese names.

## Materials and methods

Three gender prediction tools: 1) Genderize; 2) Gender API; and 3) the gender R package, were tested on a gold-standard registry of cancer clinical trialists with manually determined binary gender. Trialists' names and affiliations were sourced from the HemOnc knowledge base, a continually growing resource created to capture the standard-of-care treatments in the fields of hematology and oncology. The methodology for building the HemOnc knowledge-base has been previously described [17]. Likewise, the edibility criteria for including trialists in the HemOnc knowledgebase is defined on HemOnc.org [18]. The binary gender classifications used in our study refer to socially constructed gender categories, not biological sex [19,20]. Names in HemOnc are primarily sourced from the MEDLINE records of published clinical trials and undergo extensive normalization to account for the presence of diacritics, middle initials, misspellings, multipart last names represented as middle names, and other variations. When first names are not available through MEDLINE, the original manuscripts are examined for this information. Binary gender is determined by a combination of automated mappings of typically masculine or feminine forenames (e.g., John; Rebecca), web searches of publicly available information such as biographies on academic web pages, and consensus determinations including consultation with native speakers. If gender cannot be determined after these efforts, the author is labeled as "unknown gender." A subset of journals does not provide forenames; in these cases, the gender is labeled as "could not be determined." Country affiliations sourced from MEDLINE also undergo extensive normalization.

### Evaluation metrics

Gender prediction accuracy was defined as the percent of individuals whose gender was correctly predicted, as compared to the gold standard dataset. The percent of incorrect gender predictions and the percent of names with no predicted gender were also calculated. For binary statistical tests, gender predictions were categorized as successes or failures–correct gender predictions were defined as successes, while names with incorrect or absent predictions were failures.

### Evaluation protocol

All trialists with a gender determination were evaluated with Genderize and Gender API on 2023-11-21 using the R package httr (version 1.4.7). Both US Social Security Administration (SSA) and US Census Integrated Public Use Microdata Series (IPUMS) name datasets were used as a reference when predicting names with the gender R package [13] (version 0.6.0).

Genderize and Gender API were used to predict names with and without supplying a country of origin for the subset of authors with a singular country of affiliation. The Gender R package was only tested without supplying country names because the SSA and IPUMS methods do not provide that functionality. The SSA and IPUMS methods source names from 1932 to 2012 and 1789 to 1930 respectively [21]. The gender R package provides gender probabilities rather than explicit gender predictions. We assigned gender based on which gender was reported to have the highest probability in the R package. For example, the name Mark was classified as a man's name because the gender R package returned a 99.7% probability that 'Mark' is a man when the IPUMS method is selected.

Two-part names were concatenated without any delimiter e.g. Jean-Pierre was converted to jeanpierre. Middle names were removed, unless an author had a first initial/middle name, in

which case their middle name was used. Gender bias in name prediction was descriptively evaluated by calculating the percent of names that were misgendered, compared to the gold standard labeled dataset. In an additional analysis, accuracy differences resulting from delimiting two-part first names with different characters were evaluated. Two-part first name prediction accuracy was also evaluated using the first half of two-part names only. For example, the name Jean-Pierre was tested four ways: 1) jean-pierre; 2) jean pierre; 3) jeanpierre; and 4) jean.

In addition to predicting the gender of a first name, Genderize and Gender API also report an estimated probability that a gender prediction is correct. We evaluated the correlation between these API-reported probability estimates and the gold standard labeled dataset with linear regressions and Brier scores. Names with a reported probability less than or equal to 50% were excluded from the regression and Brier scores.

Precision, recall, and both gender-specific and global F1 scores were calculated for each gender inference tool. For gender-specific F1 scores, one gender was designated the positive prediction, and the other gender the negative prediction. Gender-specific F1 scores were calculated for both men and women. For global F1 scores, correctly gendered Men's names and correctly gendered women's names were classified as true positives. Names that yielded no gender predictions were classified as false negatives.

## Reanalysis of prior studies' data

The gender prediction accuracies of Genderize and Gender API were also separately evaluated using publicly available datasets from two studies [16,22] that tested gender prediction in 2018 and 2021, respectively. The dataset [23] provided by Santamaria 2018 consisted of 5,779 names sourced from various other datasets. The dataset [24] sourced from Sebo 2021 consisted of 6,131 Swiss physicians. The names from these public datasets were not modified prior to our evaluation on 2023-11-07. Nor were nationalities supplied to Genderize and Gender API when evaluating these public datasets, following the original experimental design.

## Software

All software accuracy comparisons were computed in R version 4.3.1. Differences in accuracy between methods were evaluated using the default R stats package implementation of McNemar's test [25]. Data analysis was facilitated with tidyverse [26] (version 2.0.0), haven [27] (version 2.5.3), readxl [28] (version 1.4.3), testthat [29] (version 3.1.10), ggpmisc [30] (version 0.5.5), and patchwork [31] (version 1.1.3) R libraries.

## Results

Out of 40,273 unique clinical trialists present in the HemOnc KB as of 2023-11-21, 37,420 (92.9%) had a resolvable name and were thus eligible for gender determination (S1 Fig). This group was sourced from 7,473 clinical trial manuscripts published between 1947–2023. After excluding trialists with gender not yet determined (n = 4,360, 11.7%), those with a determined unknown gender (n = 78, 0.2%), and those with a determined gender but initial-only first names (n = 14, <0.1%), the final analysis set included 32,968 trialists with predetermined binary gender. Of the 32,968 trialists, 11,398 (34.6%) were designated as women. There were 7849 unique names after normalizing first initial/middle name combinations to only include a middle name. The remainder of names were shared by more than one individual. Michael was the most common name, with 473 (1.4%) occurrences. Only 1,899 (24.2%) of names occurred more than twice.

Of 25,240 trialists with a known site affiliation, 24,930 (98.8%) were affiliated with sites in a single country and were assigned to the country of their affiliated institution when querying

Genderize and Gender API with nationalities. When excluding clinical trialists without a recorded country of origin, the number of trialists and unique names was 24,930 and 6,756, respectively. The final analysis set included trialists from 87 countries, the most abundant being the US with 9,485 (38%) affiliated trialists. There were 7,569 first name-country combinations that occurred only once. The most common first-name-country combination was David-US with 201 (0.8%) instances. Only 1,760 (7.1%) of first name-country combinations appeared more than twice.

All names that were misgendered more than once are catalogued in S1 through S6 Tables. The name Andrea was most frequently misgendered when calling the Genderize and Gender API services without countries in the API call. Jan was the most misgendered name for the gender R package SSA method and Genderize when countries were included. The most frequently misgendered names for Gender API when countries are included and the gender R package IPUMS method were Laurence and Nicole respectively. The 100 most common trialist name-country combinations are presented in S7 Table.

## Gender prediction accuracy when country of origin was not supplied (baseline case)

The overall accuracy of Genderize when predicting gender for the full dataset without supplying country was 96.6% with 2.3% incorrect gender predictions and 1.1% of names yielding no prediction (Table 1). Similarly, the overall accuracy of Gender API was 96.1% with 2.7% incorrect gender predictions and 1.1% of names resulting in no prediction. The accuracy of the gender R package's predictions was lower, with 79.8% and 85.7% accuracy with the IPUMS and SSA methods, respectively. Names of men were misgendered as women less than 3% of the time for all gender prediction tools (Table 1). Names of women were misgendered over 3% of the time for all services except the gender R package when using SSA data as a reference. The difference in the percent of correct gender predictions between Genderize and Gender API was significant in favor of Genderize (p<0.001). Likewise, the accuracy difference between the gender R package methods were also significant (p<0.001), in favor of the SSA method. The accuracy differences between the gender R package methods and both Genderize and Gender API were significant, in favor of Genderize and Gender API in all cases (p<0.001). Gender API demonstrated slightly higher gender prediction accuracy when two-part names were delimited with a space: the percent of correctly inferred genders rose from 96.1% to 96.3%. Precision, Recall, and F1 scores for all gender inference tools are presented in S8 Table.

**Table 1. Accuracy of gender predictions on 32,968 included trialists.**

| Method[a] | Correct, n (%) | Incorrect, n (%) | No Predictions, n (%) | Men Incorrectly Gendered as Women, n (%)[b] | Women Incorrectly Gendered as Men, n (%)[b] |
|---|---|---|---|---|---|
| Genderize | 31,857/32,968 (96.6%) | 763/32,968 (2.3%) | 348/32,968 (1.1%) | 401/21,324 (1.9%) | 362/11,296 (3.2%) |
| Gender API | 31,690/32,968 (96.1%) | 899/32,968 (2.7%) | 379/32,968 (1.1%) | 393/21,320 (1.8%) | 506/11,269 (4.5%) |
| gender (IPUMS) | 26,294/32,968 (79.8%) | 1366/32,968 (4.1%) | 5308/32,968 (16.1%) | 508/17,941 (2.8%) | 858/9719 (8.8%) |
| gender (SSA) | 28,266/32,968 (85.7%) | 590/32,968 (1.8%) | 4112/32,968 (12.5%) | 489/18,595 (2.6%) | 101/10,261 (1%) |

[a]Two-part first names were appended together without a delimiting character.

[b]Denominators are not consistent across rows because names that did not return a gender prediction for a given service were excluded.

After restricting Genderize's predictions to trialists affiliated with a single country, the percentage of correct, incorrect, and missing predictions were 96.2%, 2.6%, and 1.2% respectively (Fig 1A). Genderize achieved the highest accuracy when evaluating first names from German authors, and the lowest accuracy when evaluating names from South Korean, Chinese, Singaporean, and Taiwanese authors. When evaluating the same 24,929 clinical trialists with Gender API, the percentage of correct, incorrect, and missing predictions were 95.8%, 3%, and 1.3% respectively. Gender API also had high accuracy when predicting the gender of German authors, and the lowest accuracy when evaluating names from South Korean, Chinese, Singaporean, and Taiwanese authors. The difference in accuracy between Genderize and Gender API is significant (p<0.001), in favor of Genderize.

## Gender prediction accuracy when country of origin was supplied to the API

The gender prediction accuracies when countries of origin were supplied to Genderize and Gender API are visualized in Fig 1B. Supplying the countries of origin alongside first names in the API call decreased the percentage of correct gender predictions when using Genderize from 96.2% to 95.4%, while also reducing the percentage of incorrect predictions from 2.6% to 2.1%. Conversely, including countries of origin increased the ratio of correct gender predictions of Gender API from 95.8% to 96% and decreased incorrect predictions from 3% to 2.7%. Supplying countries also increased the percentage of names with no gender prediction for Genderize from 1.2% to 2.5%, while Gender API remained constant at 1.3%. The difference in

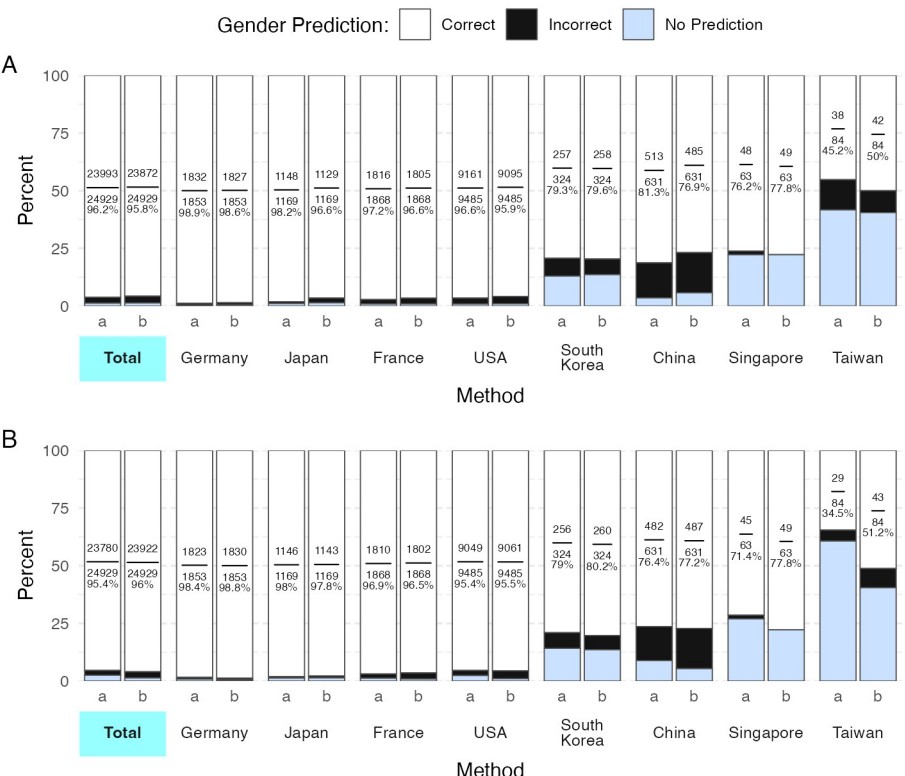

**Fig 1. Accuracy of Genderize and Gender API.** Panel A shows the gender prediction accuracies when countries are not included in the API call. Panel B shows the results when countries are included in the API call. The top 3 countries with the most trialists and the top 4 East-Asian/Southeast-Asian countries are plotted. Method a is genderize and method b is Gender API. Each bar is labeled with the fraction and percentage of correct gender predictions. Two-part first names were appended together without a delimiting character.

accuracy between Genderize and Gender API was significant in favor of Gender API (p<0.001).

## Gender prediction accuracy when using different characters to delimit two-part forenames

24,930 of the 32,968 (75.6%) trialists in our dataset were affiliated with sites in a single country and were assigned to the country of their affiliation when querying Genderize and Gender API with nationalities. Gender prediction accuracy when evaluating two-part names was higher when countries were not included in the API call in all contexts except when calling Genderize with the first half of a two-part name, e.g., Jean-Pierre as jean. Genderize was most accurate (76.4%) when no character was used to delimit two-part names, e.g., Jean-Pierre represented as jeanpierre (Fig 2). Genderize provided zero predictions for two-part first names delimited with a space. In contrast, Gender API achieved the highest gender prediction accuracy when delimiting two-part names with a space (83.5%). Gender prediction accuracy for two-part names was worse than for one-part names when countries were not included in the API call and two-part names were separated without a delimiter: OR 0.07 (95% CI 0.06–0.08) for Genderize and OR 0.08 (95% CI 0.07–0.09) for Gender API, respectively (Fig 3).

The accuracy of Genderize and Gender API were evaluated for statistical significance by comparing the percent of correct gender predictions between delimiter categories. The difference in Gender prediction accuracy between Genderize and Gender API when evaluating two-part names without a delimiting character and including countries in the API call was not significant. All other comparisons between Genderize and Gender API were significant in favor of Gender API (p<0.001).

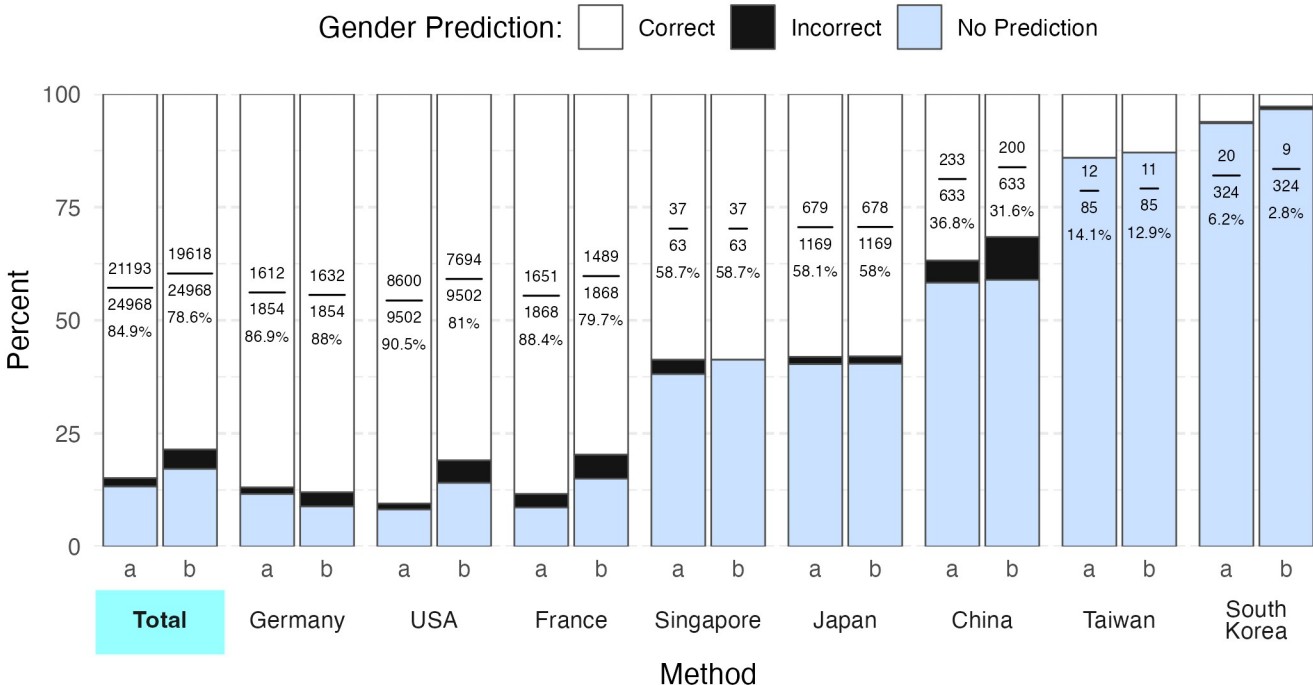

**Fig 2. Accuracy of the gender R package.** Columns a and b show the accuracy of the gender R package's SSA and IPUMS methods respectively. The top 3 countries with the most trialists and the top 4 East-Asian/Southeast-Asian countries are plotted. Only trialists affiliated with sites in a single country are plotted.

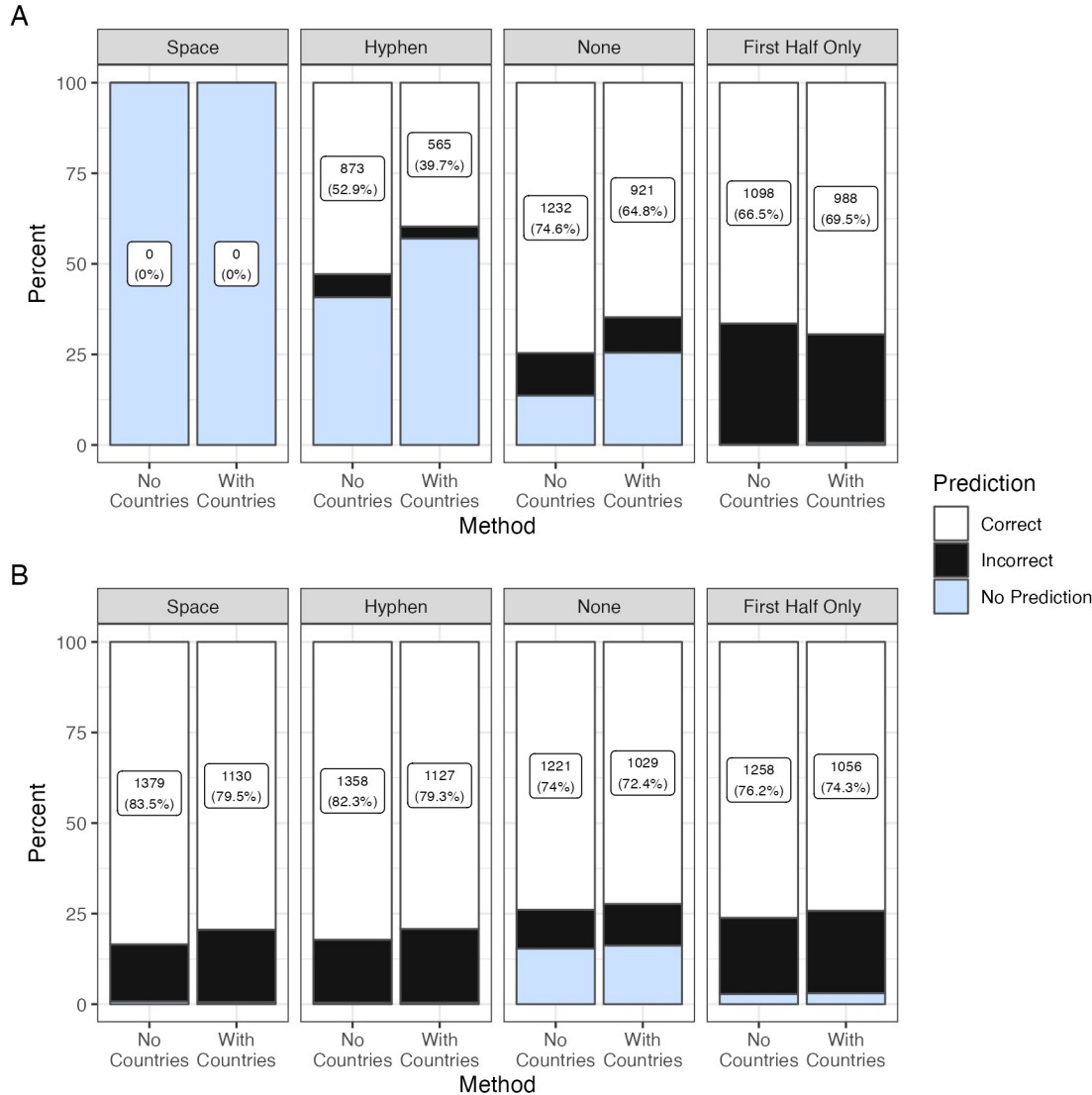

**Fig 3. Accuracy of gender predictions based on delimiter between two-part names.** Panel A is Genderize and Panel B is Gender API. Plot facets correspond to the type of delimiter separating two-part names. Stacked bars correspond to correct, incorrect, and no predictions respectively. Bars are labeled with the count and percent of correct gender predictions.

## Gender prediction accuracy by API-reported confidence thresholds

There was high agreement overall between gender prediction services and the gold standard labeled dataset (Fig 4). Genderize reported over 50% confidence in gender predictions for 32,573 (98.8%) trialists. Similarly, Gender API reported over 50% confidence for 32,587 (98.8%) trialists. Gender API demonstrated a correlation of 0.91 between its reported confidence and actual accuracy, compared to Genderize's correlation of 0.82. The Brier scores for Gender API and Genderize were 0.0077 and 0.0048 respectively.

## Replication of analyses by Santamaria 2018 and Sebo 2021

The original dataset used by Santamaria consisted of 5779 first names with known genders, 34% of whom were women. Only 0.4% of the 5779 had diacritical marks. In addition, 1.1%

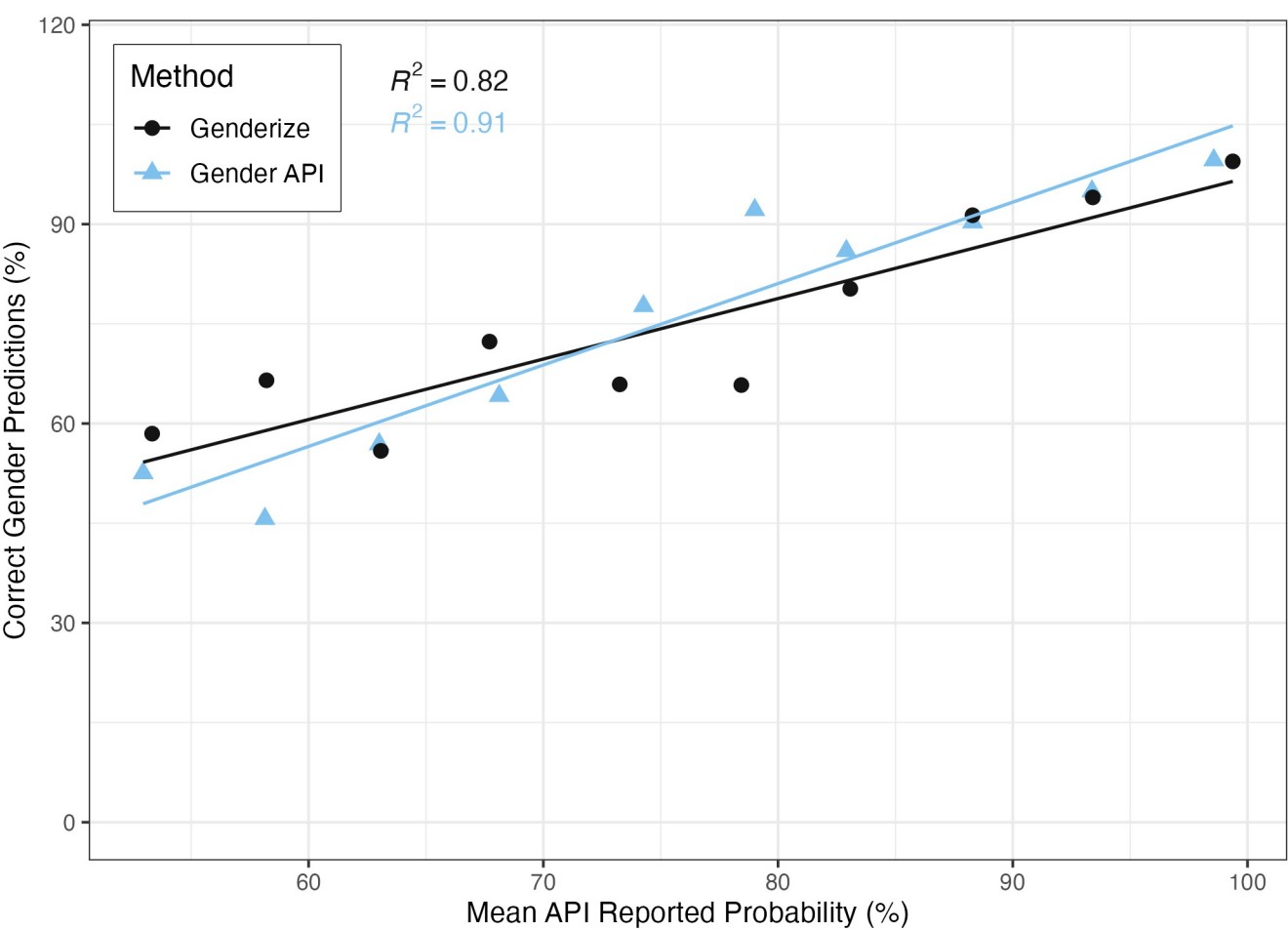

**Fig 4. Experimental name prediction accuracy at different API probability cutoffs.** Names with gender predictions were aggregated into the following API reported probability bins: 50%-55%, 55%-60%, 60%-65%, 65%-70%, 70%-75%, 75%-80%, 80%-85%, 85%-90%, 90%-95%, 95%-100%. The API reported probabilities within each bin were averaged and plotted on the x-axis. The experimentally determined gender prediction accuracies for the names in each bin are visualized on the y-axis.

and 2% of names contain spaces or hyphens, respectively. The original paper reported 80% accuracy using Genderize and 87% using Gender API. In our re-analysis, Genderize predicted the correct gender 92.5% of the time. Similarly, Gender API achieved 92.8% accuracy. The difference in accuracy between Genderize and Gender API was not statistically significant.

The dataset originally analyzed by Sebo 2021 included 6131 names of whom 50.3% were women. Diacritical marks were present in 6.6% of names. 10.2% of names contained spaces, and 6.6% of names included a hyphen. The original paper reported 81% accuracy using Genderize and 97% with Gender API. In our re-analysis, the accuracy of Genderize and Gender API on these 6131 names were 86.2% and 98% respectively. McNemar's test indicated that the differences in accuracy was statistically significant (p<0.001), in favor of Gender API. Gender API was 99.5% accurate when evaluating names with diacritical marks, while Genderize was 71.7% accurate.

## Cost and accessibility

Genderize and Gender API provide a graphical user interface, while the gender R package requires programming. Genderize [11] provides 1,000 free predictions per day, whereas

Gender API [12] only allows 100 free predictions per month. Gender API currently costs 4.85x more than Genderize for a monthly subscription that provides 100,000 predictions. The costs mentioned herein are accurate as of April 2024.

## Discussion

Tools [23,24] that infer gender from first names are commonly used by meta researchers to evaluate gender ratios in academia. A study [32] by Holman et al in 2018 reported persistent gender disparities in traditionally male-dominated fields. Gender inference technology like Genderize and Gender API facilitate programmatic gender mapping for methodologies that evaluate gender ratios based on first names.

Genderize and Gender API both demonstrated over 95% overall accuracy on our gold-standard dataset of cancer clinical trialists. Genderize was slightly more accurate than Gender API when countries were not included in the API call. Conversely, Gender API performed slightly better than Genderize when countries were included. For both services, including countries reduced the number of incorrect gender assignments at the cost of increasing the number of names with no predicted gender (Fig 1, S9 and S10 Tables). The gender R package performed worse than Genderize or Gender API (Table 1). The high accuracy of Genderize and Gender API, coupled with relatively modest usage fees, suggest that these proprietary tools may be superior to the gender R package for many use cases.

Genderize and Gender API differed in how their accuracy was affected by the delimiter separating two-part first names. Genderize was most accurate when two-part first names were appended together without a delimiter (Fig 2). In fact, Genderize appeared to be incompatible with two-part first names that were delimited by a space as the service yielded zero correct predictions when evaluating such names. Conversely, Gender API performed best when two-part first names were delimited with a space. The slightly higher overall gender prediction accuracy attained by Genderize compared to Gender API is partially an artifact of our decision to append two-part names without a delimiter in the baseline comparison, since Gender API performed best when two-part names were delimited with a space.

The gender R package was less accurate than both Genderize and Gender API on all fronts (Table 1, Figs 1 and 2). Between the two gender R package methods, the SSA method proved more accurate. Interestingly, the SSA method misgendered men as women more frequently than the reverse. In contrast, all other tools evaluated in this study misgendered women as men more frequently.

The present study provides an updated accuracy benchmark for tools that infer gender from first names. Furthermore, this study adds to a growing body of literature on the accuracy of gender inference tools. Santamaria and Mihaljevic compared Gender API, gender-guesser, Genderize, NameAPI, and NamSor in 2018, and concluded that Gender API achieved superior accuracy [16]. In 2021, Sebo compared Gender API, Genderize, and NamSor, ultimately reporting that Gender API demonstrated the highest accuracy, followed by NamSor [22].

A commonality between the present study and several previous studies was the lower prediction accuracy of Genderize and Gender API when evaluating names of people in Asian countries, with the exception of Japanese names [16,22]. The higher accuracy achieved for both services in our re-analysis of Santamaria's dataset indicates that both services have improved since 2015, although Genderize improved by a larger margin. Gender API outperformed Genderize when re-analyzing Sebo's dataset largely because Gender API handled two-part names that were delimited by spaces as well as names with diacritical marks. In fact, a follow up study [15] by the same author recommended removing diacritical marks and modifying two-part names to improve the accuracy of Genderize.

Tools that infer gender from first names will never achieve perfect accuracy because many names are gender-neutral. Additionally, the gender of certain names correlates with country of origin. For example, Andrea, Gabriele, and Daniele are more likely to correspond to men if they are of Italian origin. In fact, Andrea was the most frequently misgendered name when Genderize and Gender API were called without including countries of origin. Importantly, the higher accuracy achieved after reanalyzing Sebo and Santamaria's datasets suggests that the accuracy of Genderize and Gender API is likely to increase over time as their respective companies continue to improve their products. Indeed, algorithms such as Genderize and Gender API must be periodically calibrated in order to maintain their accuracy in changing contexts [33,34].

This study's results should be interpreted with certain caveats in mind. We did not filter out recurring first names during this analysis because the count of names in real-world datasets like ours tends to follow a long-tail distribution [35]. The process for determining the "gold-standard" gender of each of the trialists relied on inference from available information. Affiliation data was missing for a substantial subset of authors, mostly due to the older practice of MEDLINE including only the affiliation of the first author; a substantial number of high-profile oncology journals (e.g., the *Journal of Clinical Oncology* and *Blood*) did not include clear 1:1 mappings for author-to-affiliation for a period of time; this issue affects at least 320 (4%) of manuscripts in the HemOnc KB. A subset of authors in the HemOnc KB have not had their gender determined (18.1%), and this subset has some important differences from the set of determined genders. Most notably, the undetermined subset has many more Asian hyphenated names (43.8% vs 3.8%) and authors with a country of affiliation including South Korea, China, Singapore, and/or Taiwan (45.6% vs 3.42%). It is thus likely that our results represent a "best-case scenario" and that automated gender mapping will become increasingly difficult as cancer clinical trials are increasingly conducted in the Asia-Pacific region [36,37]. Additionally, a researcher's nationality in our data set does not always reflect the cultural origin of their first name as some researchers immigrated to the country of their academic affiliation. Furthermore, the database of gendered researchers in the HemOnc KB is constrained by the diversity of the clinical trialist community.

All tools we evaluated were least accurate with names from Asian countries, excluding Japan. Many Asian names with an identifiable gender in their native script become a gender-neutral representation when converted to the Latin alphabet [16]. Journals should consider publishing Asian names using native characters in addition to English translations. Even so, a 2022 study found that many Chinese characters that are traditionally used in women's names have become more gender-neutral over time [38]. Similarly, a 2023 study reported that phonological characteristics that indicate the gender of the English spelling of Korean names have also become more ambiguous post-2020 [39]. The ambiguity of gender-neutral names underscores the importance for researchers to have a public presence that includes self-reported demographic information.

In this study, the gender R package was tested with the SSA and IPUMS methods because those name datasets are more recent than the gender R package's NAPP method, which does not include names post-1910. The gender R package's Kantrowitz method was also not used because potentially gender-neutral names like Alex, Jamie, and Andrea listed as having a gender of "either", without a probability of which gender is more likely.

It is important to note that Genderize, Gender API, and the gender R package assume a gender binary. However, a recent survey [40] found that 1.6% of U.S. adults identify as transgender or nonbinary. When evaluating gender ratios using gender inference tools, it is imperative that inclusivity must be considered. However, any tool that predicts the likelihood of an individual being non-binary based solely on first names would remain challenging, due in part to the

scarcity of gender-labelled datasets that include non-binary individuals. We suggest that academic journals could facilitate research on gender equity among clinical trialists by including self-reported author gender identities. Critically, methodologies that use tools that infer binary gender must acknowledge the exclusion of non-binary individuals as a limitation. Accounting for the presence of non-binary individuals may not be feasible without self-reported gender name data sets. In response, gender researchers should attempt to develop or locate datasets that are labelled with non-binary identities when conducting meta research gender disparity analyses. For example, a 2022 study evaluated the proportions of men, women, and non-binary corresponding authors in physics journals using self-reported gender [41]. Furthermore, researchers should consider prioritizing accuracy over cost when selecting gender inference tools in order to reduce the number of names that are misgendered.

Both Genderize and Gender API demonstrated high gender prediction accuracy with non-Asian names that were highly normalized without middle or last names or diacritical marks. The cost per name evaluated with Genderize is also several times cheaper than Gender API. However, Genderize loses accuracy compared to Gender API when name formatting becomes less consistent. The SSA and IPUMS methods of the gender R package were less accurate but are open-source alternatives. The results from this study provide a new benchmark for gender inference tools. Replicating the studies of Santamaria 2018 and Sebo 2021 demonstrated that Genderize and Gender API have improved over time. Accordingly we should expect that gender prediction accuracy and features of Genderize and Gender API would continue to be more accurate over time.

## Supporting information

**S1 Fig. Flowchart depicting trialist inclusions from HemOnc Knowledgebase (KB).** The HemOnc KB is a growing resource, and 4,360 trialists had not had their genders evaluated at the time of this study. Of the 32,968 trialists included in the study, 24,930 were affiliated with sites in a single country.
(DOCX)

**S1 Table. Names misgendered more than once, Gender API with country provided.**
(CSV)

**S2 Table. Names misgendered more than once, Gender API without country provided.**
(CSV)

**S3 Table. Names misgendered more than once, gender R package, IPUMS method.**
(CSV)

**S4 Table. Names misgendered more than once, gender R package, SSA method.**
(CSV)

**S5 Table. Names misgendered more than once, Genderize API with country provided.**
(CSV)

**S6 Table. Names misgendered more than once, Genderize API without country provided.**
(CSV)

**S7 Table. 100 most common trialist name-country combinations.**
(DOCX)

**S8 Table. Precision, recall, and F1 scores.**
(DOCX)

**S9 Table. Gender prediction accuracy for all countries with at least 20 trialists when countries are not included in the API call.**
(DOCX)

**S10 Table. Gender prediction accuracy for all countries with at least 20 trialists when countries are included in the API call.**
(DOCX)

## Acknowledgments

We would like to acknowledge the efforts of the editorial board of HemOnc.org.

## Author Contributions

**Conceptualization:** Alexander D. VanHelene, Ishaani Khatri, Jeremy L. Warner.

**Data curation:** Alexander D. VanHelene, Sanjay Mishra, Jeremy L. Warner.

**Formal analysis:** Alexander D. VanHelene, C. Beau Hilton, Sanjay Mishra, Jeremy L. Warner.

**Funding acquisition:** Sanjay Mishra, Jeremy L. Warner.

**Investigation:** Alexander D. VanHelene, Sanjay Mishra, Jeremy L. Warner.

**Methodology:** Alexander D. VanHelene, Sanjay Mishra, Jeremy L. Warner.

**Resources:** Jeremy L. Warner.

**Software:** Alexander D. VanHelene.

**Validation:** Alexander D. VanHelene, Jeremy L. Warner.

**Visualization:** Alexander D. VanHelene, Sanjay Mishra, Ece D. Gamsiz Uzun, Jeremy L. Warner.

**Writing – original draft:** Alexander D. VanHelene, Ishaani Khatri, Jeremy L. Warner.

**Writing – review & editing:** Alexander D. VanHelene, Ishaani Khatri, C. Beau Hilton, Sanjay Mishra, Ece D. Gamsiz Uzun, Jeremy L. Warner.

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
