## [Decision Letter · Decision Letter 0]

26 Mar 2024

PDIG-D-24-00041

Inferring Gender from First Names: Comparing the Accuracy of Genderize, Gender API, and the gender R Package on Authors of Diverse Nationality

PLOS Digital Health

Dear Dr. Warner,

Thank you for submitting your manuscript to PLOS Digital Health. After careful consideration, we feel that it has merit but does not fully meet PLOS Digital Health's publication criteria as it currently stands. Therefore, we invite you to submit a revised version of the manuscript that addresses the points raised during the review process.

Please submit your revised manuscript within 60 days May 25 2024 11:59PM. If you will need more time than this to complete your revisions, please reply to this message or contact the journal office at digitalhealth@plos.org. Please include the following items when submitting your revised manuscript:

We look forward to receiving your revised manuscript.

Kind regards,

Miguel Ángel Armengol de la Hoz, Ph.D.

Section Editor

PLOS Digital Health

Journal Requirements:

Additional Editor Comments (if provided):

Reviewers' comments:

Reviewer's Responses to Questions

**Comments to the Author**

1. Does this manuscript meet PLOS Digital Health’s publication criteria? Is the manuscript technically sound, and do the data support the conclusions? The manuscript must describe methodologically and ethically rigorous research with conclusions that are appropriately drawn based on the data presented.

Reviewer #1: Yes

Reviewer #2: Yes

Reviewer #3: Partly

Reviewer #4: Partly

Reviewer #5: Yes

Reviewer #6: Partly

Reviewer #7: Partly

2. Has the statistical analysis been performed appropriately and rigorously?

Reviewer #1: N/A

Reviewer #2: Yes

Reviewer #3: Yes

Reviewer #4: N/A

Reviewer #5: Yes

Reviewer #6: Yes

Reviewer #7: Yes

3. Have the authors made all data underlying the findings in their manuscript fully available (please refer to the Data Availability Statement at the start of the manuscript PDF file)?

Reviewer #1: Yes

Reviewer #2: Yes

Reviewer #3: Yes

Reviewer #4: Yes

Reviewer #5: Yes

Reviewer #6: Yes

Reviewer #7: No

4. Is the manuscript presented in an intelligible fashion and written in standard English?

Reviewer #1: Yes

Reviewer #2: Yes

Reviewer #3: Yes

Reviewer #4: Yes

Reviewer #5: Yes

Reviewer #6: Yes

Reviewer #7: Yes

5. Review Comments to the Author

Reviewer #1: Several researches uses such type of technology to seek assistance in their work. In light of this, it becomes a necessary point to determine which technology could be best suited to assist in the research work. Very useful topic and well presented article. Just kindly re-check once the line 280-283 of Pg-15 as it develops confusion regarding the observed result. Kindly clarify once.

Reviewer #2: The study of VanHelene et al. compared the accuracy and cost of commercial software Genderize and Gender API, and the open-source gender R package, in inferring gender from first names. It found that Genderize and Gender API had high accuracy rates, especially for German authors, but were less accurate for authors from South Korea, China, Singapore, and Taiwan. The study's merit lies in providing valuable insights into the performance and cost-effectiveness of gender inference tools, aiding researchers in making informed decisions when studying gender disparities.

Comments:

1. Abstract: incomplete sentence „The accuracy of the gender R package was only evaluated without supplying countries of origin since.“ 

2. Line 55: Why are gender gaps in the authorship (not patient inclusion) of cancer clinical trials of particular concern?

3. Lines 65-66, 80-81, 327: How did the authors define Western / Non-Western names? The present approach only evaluated names from Western / Asian countries.

4. Lines 106-110: The R package only returns aggregate percentages for each name. The authors should describe in greater detail how they calculated the numbers in Table 1 based on these aggregate percentages. Did they a) multiply probabilities with total number of persons with given name and round the result or b) assume that all persons of a given name belong to the same gender? Example: index = 53717 / first_name = ilya / probability = 0.5008 / predicted_gender = woman / man_count = 2 / woman_count = 0. In this example scenario a) would yield 1 man and 1 woman (accuracy 50%) while scenario b) would yield 0 man and 2 woman (accuracy 0%). 

5. Lines 145-151: The addition of an flow chart depicting inclusion/exclusion of trialists as described would be helpful for the reader.

6. Lines 171-173: A supplementary list of the most common names that yielded incorrect predictions would be useful for later studies.

7. Lines 174-175 misses a direct statistical comparison between all methods used. Only comparisons between Genderize vs. Gender API and SSA vs. IPUMS are reported.

8. Lines 176-177: For the sake of clarity, these lines should be moved to the result section that compares the different delimiters.

9. Line 207: Figure 1 should include bar graphs and statistical comparisons for the R package options that were investigated

10. Line 274: Please provide year and month of this cost analysis as prices can be highly dynamic.

11. What year or range of years did the authors impute into the R algorithm? This could greatly impact the accuracy of the algorithm.

Reviewer #3: This paper could benefit from a more comprehensive discussion on several aspects:

 - Why and when names-based gender identification needs to be used, including the limitations and potential biases inherent in this approach

 - An in-depth analysis of the methodological challenges, particularly concerning names that are gender-neutral, and an estimation of the maximum accuracy achievable under these conditions.

 - Identifying names that are frequently misclassified would enhance the paper's insightfulness.

 - Recommendations for more effective methods, such as integrating / combining multiple approaches or adding more contextual information to enhance prediction accuracy, should be explored.

 - The impact of including the country of origin on prediction accuracy needs further examination, including whether specific names significantly affect the results.

 - The discussion on non-binary gender options requires clarification on how they integrate with name-based gender prediction methods, questioning the assumption that certain names can indicate non-binary identities.

 - This topic should extend into a broader dialogue on methodological limitations, citing examples like Meryll, Parker, and Jamie, which are ambiguous in terms of gender.

 - Suggestions for improving accuracy, particularly in regions where current methods show diminished effectiveness, would greatly contribute to the paper's value.

 - Exploring how the accuracy of the model might shift over time, in response to changes in the popularity of names, would also be insightful.

Reviewer #4: I thoroughly enjoyed reading the article titled "Inferring Gender from First Names: Comparing the Accuracy of Genderize, Gender API, and the gender R Package on Authors of Diverse Nationality". It provided a comprehensive overview of an important subject.

This article presents a comparative analysis of the accuracy and cost-effectiveness of three gender inference tools commonly used in meta-research: Genderize, Gender API, and the gender R package. The study evaluates these tools' performance in predicting the gender of authors listed in clinical trial publications across various countries.

One of the strengths of this article is its clear objective, which is to compare the accuracy and cost of different gender inference tools. The use of a large multinational dataset also adds credibility to the findings, as it allows for a comprehensive evaluation of the tools' performance across diverse cultural contexts (generalization).

Although the manuscript follows the IMRaD format (Introduction, Method, Results and Discussion), which is a widely recognized structure used in scientific writing, especially in journals such as PLOS ONE, I missed an independent and robust literature review section.

The authors rely on a dataset of clinical trial authors sourced from the HemOnc knowledge base, which may not accurately represent the diversity of names and genders worldwide [I am not sure]. Can you please elaborate on this subject?

The article needs to address the limitations of binary gender classification and the exclusion of transgender and nonbinary individuals from the analysis. This reflects a broader issue of gender essentialism. Thus, the article treats gender as a binary variable without critically engaging with the social construction of gender and its fluidity. By reinforcing binary gender norms, the article perpetuates outdated and harmful stereotypes, neglecting the diverse experiences and identities of individuals across the gender spectrum.

The comparison between the accuracy of Genderize and Gender API appears to ignore critical specificities regarding names from non-Western cultures. While the study acknowledges the lower accuracy of these tools with Asian names, it does not provide meaningful information about the reasons behind this disparity or propose solutions to resolve it. Furthermore, the article's emphasis on cost-effectiveness ignores ethical considerations and the potential consequences of prioritizing accessibility over accuracy in gender prediction.

The article could benefit from a discussion of future research directions and potential improvements in gender prediction methodologies. It could include exploring alternative approaches such as probabilistic modeling, incorporating user feedback and community input, and integrating intersectional perspectives to address the limitations of existing tools. In essence, I believe that the article has the potential to significantly enhance its robustness in its present form. Any limitations that cannot be adequately addressed at this stage should be acknowledged, and avenues for further research should be suggested for future exploration.

In conclusion, although the article introduces new insights into the challenges and complexities of gender prediction in research, the interpretation of results and ethical considerations somewhat undermine the credibility and relevance of its conclusions. A more rigorous and inclusive approach to gender inference, along with critical reflection on the limitations of existing tools, is essential to promoting gender equity and inclusion in research practices.

Reviewer #5: Good study that is well written and communicated. Some comments are needed for improvement:

-The concluding statement of the abstract needs rephrasing to sum up the findings.

-In the methods: The list of definitions could be placed as a separate section

-Make sure all the criteria are defined in the method for e.g.high profile journals

-Discuss in detail the differences in predictions between the different tools used. This is shown in the results communicated but not discussed thoroughly 

-In the discussion, highlight the performance with non-Western names

Reviewer #6: Report

Research objective:Comparing prediction accuracies of three prediction tools;

Genderize,Gender API and open source gender R package using multinational dataset of clinicial tria author names.

Observations

 1.Research objective clearly stated

 2.Research methods stated

 3.machine learning algorithms used, stated with brief explanations

Issues

 1. How does api's work?

 2. what is the basis for comparing these apis?

 3. How does the internal structures of these apis influence these comparisons?

 4. Concerning the choice of these api's, what are their related differences?

 4. what are the study constraints and identified bias.

Reviewer #7: This research provides a helpful tool to evaluate the gender attribution. However, it is unclear what is the significance or relevance to digital health. For example, is this helping to improve the accuracy of trials? How could this improve the recognition of gender through first names? Evaluating packages/tools may carry practical value, but it needs to be highlighted. Also, the statistical method used is very basic.

6. PLOS authors have the option to publish the peer review history of their article (what does this mean?). If published, this will include your full peer review and any attached files.

**Do you want your identity to be public for this peer review?** For information about this choice, including consent withdrawal, please see our Privacy Policy.

Reviewer #1: No

Reviewer #2: No

Reviewer #3: No

Reviewer #4: No

Reviewer #5: No

Reviewer #6: No

Reviewer #7: No

---

## [Decision Letter · Decision Letter 1]

16 Jul 2024

PDIG-D-24-00041R1

Inferring Gender from First Names: Comparing the Accuracy of Genderize, Gender API, and the gender R Package on Authors of Diverse Nationality

PLOS Digital Health

Dear Dr. Warner,

Thank you for submitting your manuscript to PLOS Digital Health. After careful consideration, we feel that it has merit but does not fully meet PLOS Digital Health's publication criteria as it currently stands. Therefore, we invite you to submit a revised version of the manuscript that addresses the points raised during the review process.

Please submit your revised manuscript within 60 days Sep 14 2024 11:59PM. If you will need more time than this to complete your revisions, please reply to this message or contact the journal office at digitalhealth@plos.org. Please include the following items when submitting your revised manuscript:

We look forward to receiving your revised manuscript.

Kind regards,

Miguel Ángel Armengol de la Hoz, Ph.D.

Section Editor

PLOS Digital Health

Journal Requirements:

Additional Editor Comments (if provided):

Reviewers' comments:

Reviewer's Responses to Questions

**Comments to the Author**

1. If the authors have adequately addressed your comments raised in a previous round of review and you feel that this manuscript is now acceptable for publication, you may indicate that here to bypass the “Comments to the Author” section, enter your conflict of interest statement in the “Confidential to Editor” section, and submit your "Accept" recommendation.

Reviewer #8: (No Response)

Reviewer #9: All comments have been addressed

Reviewer #10: All comments have been addressed

Reviewer #11: All comments have been addressed

2. Does this manuscript meet PLOS Digital Health’s publication criteria? Is the manuscript technically sound, and do the data support the conclusions? The manuscript must describe methodologically and ethically rigorous research with conclusions that are appropriately drawn based on the data presented.

Reviewer #8: No

Reviewer #9: Yes

Reviewer #10: Partly

Reviewer #11: Yes

3. Has the statistical analysis been performed appropriately and rigorously?

Reviewer #8: No

Reviewer #9: Yes

Reviewer #10: No

Reviewer #11: Yes

4. Have the authors made all data underlying the findings in their manuscript fully available (please refer to the Data Availability Statement at the start of the manuscript PDF file)?

Reviewer #8: No

Reviewer #9: Yes

Reviewer #10: (No Response)

Reviewer #11: Yes

5. Is the manuscript presented in an intelligible fashion and written in standard English?

Reviewer #8: No

Reviewer #9: Yes

Reviewer #10: Yes

Reviewer #11: Yes

6. Review Comments to the Author

Reviewer #8: COMMENTS NOT ADDRESSED FULL

Reviewer #9: Thanks to the authors for answering my concerns. The authors had addressed my concerns. Thus, I recommend the journal accept the current manuscript.

Reviewer #10: Revisions Suggested :- 

1. Accuracy alone can be misleading, especially in datasets with class imbalance. Precision, recall, and F1 score provide a more detailed understanding of the performance by considering the trade-offs between false positives and false negatives. 

2. The authors have noted the presence of missing values present in the data, Trialists with unresolved names or undetermined genders were excluded from the analysis. The authors should provide a detailed explanation of how this exclusion might affect the results. The exclusion of a significant proportion of trialists with missing data (7.1% unresolved names, 11.9% unresolved genders) could introduce bias, especially if the missing data is not random.

3. The introduction should briefly discuss the importance of accurate gender prediction in broader contexts beyond academia, such as implications for public health, policy-making, and diversity metrics.

4. The manuscript should provide a more in-depth analysis of why non-Western names, especially from Asian countries, are less accurately predicted. This should include a discussion of cultural and linguistic factors affecting name-gender associations.

5. Expand the discussion on the limitations of using binary gender classifications and the exclusion of non-binary and transgender identities. This section should also discuss the potential biases introduced by these limitations and suggest ways future research can address them.

Reviewer #11: I think this paper is well-written and transparent, with an appropriate statistical analysis and that its conclusions follow from the analysis. The figures and tables are all appropriate. 

However, I think the paper has a fundamental flaw that makes it of limited usefulness which I will try to explain here.

From examination of the Genderize and Gender API websites, as well as the source code of the gender package (and the genderdata package on which it depends), it appears that all three "methods" discussed in the paper are actually the same method: a simple look-up of the given name on the database associated with the method. Of course, there is some simple text-parsing with the API-based tools that allows the input to be less sensitive to formatting (in particular with Gender API), but otherwise this study is comparing the accuracy of four databases, not three methods. The R package actually allows the user to access the Genderize database by specifying method = "genderize" rather than method = "ssa" or method = "ipums" inside the gender() function, and this would presumably give identical results to those reported for Genderize in this paper. The R package also contains two other historical databases, namely the North American Population Project (NAPP) database, and the Kantrowitz name corpus database. Neither are even mentioned in this paper for some reason.

That being the case, it seems the paper is actually comparing four databases: the actively commercially maintained Gender API database, the actively commercially maintained Genderize database, the historical, unmaintained SSA database and the even older unmaintained IPUMS database.

The paper's conclusion, namely that Genderize and Gender API are more accurate than the gender R package, is saying little other than actively maintained and up-to-date name databases are more accurate than unmaintained historical databases that have not been updated for many years. This should surprise no-one, yet that is not how the results are presented.

Because the paper treats the APIs like a "black box", the casual reader would be left with the impression that they are something more sophisticated than they actually are, which is simply well-maintained lookup tables. Furthermore, the R package can presumably be induced to equalling their accuracy with a simple change of input parameters and an API key for the Genderize database, so framing this as the R package being less accurate is just misleading. It is the open source data to which the package has free access that is lacking, not the software itself.

In addition, I have concern about the longevity of any conclusions drawn from this study. APIs change, companies go bust, R packages have a fairly high attrition rate too (maybe 40% per 10 years). This paper describes a snapshot of three currently available methods to predict gender from names that are very unlikely to all stay in their current format for more than few years, particularly with the advent of context-rich AI analysis. 

On that subject, it is entirely possible that even the current iteration of ChatGPT could give a more accurate gender prediction than any of the methods discussed here, and this will only improve with time. Ultimately, a medical / sociological scientific report on an the accuracy of inscrutable proprietry method that is subject to change and cannot be replicated does little to advance our knowledge.

7. PLOS authors have the option to publish the peer review history of their article (what does this mean?). If published, this will include your full peer review and any attached files.

**Do you want your identity to be public for this peer review?** For information about this choice, including consent withdrawal, please see our Privacy Policy. 

Reviewer #8: Yes: Mubashir Zafar

Reviewer #9: No

Reviewer #10: No

Reviewer #11: No

---

## [Decision Letter · Decision Letter 2]

9 Sep 2024

Inferring Gender from First Names: Comparing the Accuracy of Genderize, Gender API, and the gender R Package on Authors of Diverse Nationality

PDIG-D-24-00041R2

Dear Dr. Warner,

We are pleased to inform you that your manuscript 'Inferring Gender from First Names: Comparing the Accuracy of Genderize, Gender API, and the gender R Package on Authors of Diverse Nationality' has been provisionally accepted for publication in PLOS Digital Health.

Best regards,

Miguel Ángel Armengol de la Hoz, Ph.D.

Section Editor

PLOS Digital Health

Reviewer Comments (if any, and for reference):

Reviewer's Responses to Questions

**Comments to the Author**

1. If the authors have adequately addressed your comments raised in a previous round of review and you feel that this manuscript is now acceptable for publication, you may indicate that here to bypass the “Comments to the Author” section, enter your conflict of interest statement in the “Confidential to Editor” section, and submit your "Accept" recommendation.

Reviewer #8: (No Response)

Reviewer #9: All comments have been addressed

Reviewer #10: All comments have been addressed

Reviewer #11: All comments have been addressed

2. Does this manuscript meet PLOS Digital Health’s publication criteria? Is the manuscript technically sound, and do the data support the conclusions? The manuscript must describe methodologically and ethically rigorous research with conclusions that are appropriately drawn based on the data presented.

Reviewer #8: Partly

Reviewer #9: Yes

Reviewer #10: Yes

Reviewer #11: Yes

3. Has the statistical analysis been performed appropriately and rigorously?

Reviewer #8: No

Reviewer #9: Yes

Reviewer #10: Yes

Reviewer #11: Yes

4. Have the authors made all data underlying the findings in their manuscript fully available (please refer to the Data Availability Statement at the start of the manuscript PDF file)?

Reviewer #8: Yes

Reviewer #9: Yes

Reviewer #10: Yes

Reviewer #11: Yes

5. Is the manuscript presented in an intelligible fashion and written in standard English?

Reviewer #8: Yes

Reviewer #9: Yes

Reviewer #10: Yes

Reviewer #11: Yes

6. Review Comments to the Author

Reviewer #8: method section need to rewrite

missing some information , validity of questionoare, sample size calculation, appropriate statistical test

Reviewer #9: The authors have addressed my all concerns.

Reviewer #10: All my comments and concerns have been addressed.

Reviewer #11: Thank you for taking the time to respond to my comments and updating the paper accordingly. While I think my concerns about the framing of the results (and the longevity of their validity) are justified, and while I do think that the peer-review process is an important part of assessing the likely utility of a paper, I accept the authors' point that it will be useful for meta researchers to be able to cite the accuracy of existing tools in this field. On reflection, I would be happy to cite this paper as justification for selecting one of the tools. The authors are to be commended for being so accommodating and responsive to reviewer feedback.

7. PLOS authors have the option to publish the peer review history of their article (what does this mean?). If published, this will include your full peer review and any attached files.

**Do you want your identity to be public for this peer review?** For information about this choice, including consent withdrawal, please see our Privacy Policy.

Reviewer #8: **Yes: **Mubashir Zafar

Reviewer #9: No

Reviewer #10: **Yes: **Aasim Ayaz Wani

Reviewer #11: No
